# On the Connections between TRPM Channels and SOCE

**DOI:** 10.3390/cells11071190

**Published:** 2022-04-01

**Authors:** Guilherme H. Souza Bomfim, Barbara A. Niemeyer, Rodrigo S. Lacruz, Annette Lis

**Affiliations:** 1Department of Molecular Pathobiology, New York University College of Dentistry, New York, NY 10010, USA; ghs5@nyu.edu; 2Department of Molecular Biophysics, Center for Integrative Physiology and Molecular Medicine, School of Medicine, Saarland University, 66421 Homburg, Germany; barbara.niemeyer@uks.eu; 3Department of Biophysics, Center for Integrative Physiology and Molecular Medicine, School of Medicine, Saarland University, 66421 Homburg, Germany

**Keywords:** Ca^2+^ signaling, TRPM channels, SOCE, ORAI channels

## Abstract

Plasma membrane protein channels provide a passageway for ions to access the intracellular milieu. Rapid entry of calcium ions into cells is controlled mostly by ion channels, while Ca^2+^-ATPases and Ca^2+^ exchangers ensure that cytosolic Ca^2+^ levels ([Ca^2+^]_cyt_) are maintained at low (~100 nM) concentrations. Some channels, such as the Ca^2+^-release-activated Ca^2+^ (CRAC) channels and voltage-dependent Ca^2+^ channels (CACNAs), are highly Ca^2+^-selective, while others, including the Transient Receptor Potential Melastatin (TRPM) family, have broader selectivity and are mostly permeable to monovalent and divalent cations. Activation of CRAC channels involves the coupling between ORAI1-3 channels with the endoplasmic reticulum (ER) located Ca^2+^ store sensor, Stromal Interaction Molecules 1-2 (STIM1/2), a pathway also termed store-operated Ca^2+^ entry (SOCE). The TRPM family is formed by 8 members (TRPM1-8) permeable to Mg^2+^, Ca^2+^, Zn^2+^ and Na^+^ cations, and is activated by multiple stimuli. Recent studies indicated that SOCE and TRPM structure-function are interlinked in some instances, although the molecular details of this interaction are only emerging. Here we review the role of TRPM and SOCE in Ca^2+^ handling and highlight the available evidence for this interaction.

## 1. Introduction

Cation-conducting channel proteins in the plasma membrane play important roles in a multitude of cellular processes [1]. Most of these channels experience conformational changes from closed to open states allowing the passage of thousands of ions in response to chemical or mechanical signals [2]. Although these channels provide selective permeability to cations over anions determined by the amino acids lining the pore and the pore diameter, they can adopt multiple roles in cell signaling due to the variable selectivity for cations [2]. This is the case for the non-selective transient receptor potential melastatin (TRPM) family, which is capable of conducting monovalent (Na^+^ and K^+^) and divalent (Mg^2+^ and Ca^2+^) cations [3]. Less selective cation channels of the TRPC family (i.e., review by Saul and Hoth [4]) and of other families also contribute, but are not included in the present overview. By contrast, the specialized Ca^2+^ release-activated Ca^2+^ (CRAC) channels, mediating store-operated Ca^2+^ entry (SOCE), which are generated by the ORAI1-3 proteins, are a 1000-fold more selective for Ca^2+^ than Na^+^ ions [5]. Recent studies indicate that the TRPM and the CRAC channels interact in some expected ways but also by novel mechanisms [6,7]. Here we specifically review the role of TRPM channels in Ca^2+^ homeostasis and highlight recent advances toward understanding the potential synergy between TRPM and ORAI channels.

### 1.1. General Features of TRPM Channels

TRPM channels are a subfamily of the TRP superfamily composed of eight members denoted as TRPM1 to TRPM8 and grouped based on sequence homology as follows: (1) TRPM1 and TRPM3; (2) TRPM2 and TRPM8; (3) TRPM4 and TRPM5; and (4) TRPM6 and TRPM7 [3]. With the exception of the monovalent selective TRPM4 and TRPM5 channels, the remainder of the proteins in the family are non-selective cation channels that participate in a heterogeneous range of physiological processes including temperature and redox sensing, light sensing, embryonic development and Mg^2+^ homeostasis [3,8,9]. Although TRPM4 and TRPM5 are Ca^2+^ activated channels, they are impermeable to Ca^2+^ [3,10,11]. The general structural architecture of TRP channels consists of six transmembrane helical domains (TM1-TM6), with a loop between TM5 and TM6 forming the channel pore, and N- and C-terminal regions located in the cytosol [3,11]. Each TRPM subunit has a ~850 amino acids cytosolic domain, making TRPM members the largest proteins of the TRP superfamily [9,10]. The ion channel regions of TRPM2, TRPM6 and TRPM7 are linked to an intrinsic enzymatic domain within the C-terminus [12,13]. TRPM6 and TRPM7 contain serine/threonine α-kinase domains, while TRPM2 channel has a NUDT9-H domain [12,14]. These chanzymes modulate cellular functions either by inward ion currents through the pore and/or by phosphorylating downstream proteins via its enzymatic domain [3,10]. Table 1 lists features of the TRPM channels including their ion permeability and activation. Mutations in the genes encoding TRPM channels result in channelopathies including cancer [15], neuropathic pain [16], inflammation [17], hypertension [18,19], diabetes [20] and hypomineralization [21,22]. The TRPM channel pore domain located between TM5-6, surrounded by the S1-S4 domain and connected through the S4-S5 portion, appear to play an important role in channel gating [3,10]. All members of the TRPM family have a conserved Ca^2+^ binding site, but available data show that this binding site is important for the gating of only TRPM2 [23] and TRPM8 [24]. Several studies have indicated that TRPM channels are expressed in the membranes of intracellular organelles in addition to their plasmalemmal localization [3,6,25], but their functions in organelles are not completely understood and will not be discussed here.

### 1.2. General Features of the CRAC Channels

CRAC channels are formed by the endoplasmic reticulum (ER) resident Ca^2+^ sensors stromal interaction molecules 1-2 (STIM1/2) [26,27], and by the highly Ca^2+^-selective conductance pore ORAI1-3 proteins [28,29], forming the dominant store-operated Ca^2+^ entry denominated as SOCE [30,31,32]. The most common activation mechanism of SOCE involves the binding of a ligand to receptors in the plasma membrane (PM) which then couple and activate phospholipase C enzymes (PLC) to produce inositol 1,4,5 triphosphate (IP_3_) and diacylglycerol (DAG). The ensuing binding of IP_3_ to its receptor in the ER membrane results in a rapid decline in ER Ca^2+^ concentration, which subsequently causes oligomerization, migration of STIM1/2 to ER-PM contact sites, where the unfolded C-terminal CAD/SOAR domains trap and couple to the ORAI1-3 channels activating SOCE [5,33,34]. The hetero multimerization between STIM1 and STIM2 and different ORAI1-3 subunits result in distinct SOCE and CRAC biophysical properties [35,36,37,38,39]. Reports showed that loss-of-function mutations or knockdown of *ORAI2* and *ORAI3* genes results in an enhancement of SOCE [33,40,41,42,43]. Also, ORAI3 tunes-down efficient STIM1 gating when in a heteromeric complex with ORAI1 channels [44].

There are several well-known pharmacological inhibitors of SOCE and a major challenge has been to identify molecules with suitable characteristics (i.e., potency, selectivity, toxicology) to offer positive clinical applications [32,45]. The first generation of SOCE inhibitors such as MRS-1845, lanthanides (Gd^3+^/La^3+^), imidazoles (SKF-96365) and 2-APB, helped define basic SOCE cellular properties [46], although their poor selectivity significantly limited their use [32]. Newer SOCE inhibitors including compounds such as synta-66, BTP2 (YM-58483), RO2959, AnCoA4 and GSK-7975A have shown improved pharmacological characteristics in terms of potency and selectivity [46,47]. Synta-66, CM4620, RO2959 and AnCoA4 appear to be effective tools showing no significant off-target effects on TRP or voltage-gated channels [32,48,49]. The in vivo potency, efficacy and selectivity of these SOCE inhibitors remain less explored, but recent studies indicate that several new compounds are now in clinical trials [32,47,50,51].

## 2. TRPM1, TRPM2 and TRPM3 Channels

The first member of the TRPM family cloned and identified was TRPM1 in 1998 [52]. TRPM1 channels are involved in photoresponses in retinal cells in *Drosophila* and in mice [53,54]. In humans, TRPM1 is also associated with skin pigmentation and homozygous loss of *TRPM1* results in retinal blindness [53,55]. Although TRPM1 and TRPM3 channels share ~70% sequence homology, their functions are quite different with TRPM3 acting as a thermoreceptor in detecting noxious hot (~40 °C) temperatures and heat-associated inflammation [54,56]. In addition to Ca^2+^, TRPM1 is permeable to other divalent ions such as Mn^2+^ and Mg^2+^, while TRPM3 has permeability to both monovalent and divalent cations [57]. The ionic conductance of TRPM1 channels is ~76 pS and ~65 to 130 pS for TRPM3 [53,57,58].

The activation mechanism of the TRPM1 channels has not been fully elucidated yet. Recent findings suggest that the intracellular uncoupling of the Gα_o_/Gβ_γ_ subunit of G-proteins, after its activation, leads to the channel’s closure [59]. Also, the activation of protein kinase C-alpha (PKCα) reduces the inhibition of TRPM1 by Mg^2+^ ions [60]. The heat compound capsaicin has been used to investigate TRPM1 function although it is a non-selective agonist [61]. TRPM3 channels can be activated by pregnenolone sulfate, CIM0216 and hypotonic solutions [56,62] that induces an increase in [Ca^2+^]_cyt_ leading to Ca^2+^/calmodulin modulation and subsequent activation of mitogen-activated protein kinases (MAPKs) [57,62]. The lack of selective agonists of TRPM3 channels is evidenced by its activation by several metabolites and synthetic and plant-derived compounds including cholesterol, mefenamic acid, and the antidiabetic PPARγ-agonists rosiglitazone and troglitazone [56,57,62]. Functional studies using HEK-293 cells overexpressing TRPM3 channels showed that flavanones reduced pregnenolone sulfate-induced [Ca^2+^]_cyt_ elevation [63]. Also, changes in ion concentration can negatively affect channel activity with increases in [Ca^2+^]_cyt_ inhibiting TRPM1 and TRPM3 channels, intracellular Zn^2+^ blocking TRPM1, and Mg^2+^ inhibiting TRPM3 channels [8,64].

TRPM2 channels, which contain an enzymatic domain, sense warm temperatures and are also associated with the inflammatory cascade [65]. For example, *Trpm2*-deficient mice are prone to *Listeria*-mediated infections [66]. These channels are also considered as oxidative stress-sensitive and Ca^2+^-permeable ion channels [65]. The biophysical features of TRPM2 include a large pore, an intracellular Ca^2+^ binding site linked to the pore and a cation non-selectivity without voltage-dependence [23]. The single-channel conductance is ~60–80 pS and the PCa/PNa permeability ratio is ~0.7–0.9, indicating that cation influx is predominantly of Na^+^ ions [67,68]. Its permeability to Ca^2+^ and Mg^2+^ is maintained by amino acid residues located between the pore helix and the selectivity filter, being regulated by ADP ribose (ADPR), reactive oxygen and nitrogen species (ROS/RNS), and the phospholipid phosphatidylinositol 4,5-bisphosphate (PIP_2_) [19,65]. Recent evidence suggests that the activation of TRPM2 channels is linked to its enzymatic domain [69,70,71,72]. Interestingly, TRPM2 is catalytically inactive in humans but not in invertebrates [72]. At least one study reported a connection between TRPM2 and SOCE, although this link is indirect [73]. Salivary glands require SOCE for fluid secretion and radiation treatment of the gland activated a TRPM2-dependent pathway involving mitochondria which caused a caspase-mediated cleavage of STIM1 and loss of SOCE [73].

## 3. TRPM4/TRPM5 Channels

Unlike other TRPM family members, TRPM4 and TRPM5 channels are unique due to their much higher permeability to monovalent cations compared to divalent ions [74,75]. Their role in altering Ca^2+^ homeostasis is therefore more indirect. However, both channels require increases in intracellular Ca^2+^ levels to become activated and are also regulated by PIP_2_. TRPM5 channels are 5 to 10-fold more sensitive to Ca^2+^ when compared to TRPM4 [74,76,77]. TRPM5 was originally identified as MTR1, and was later found to be co-expressed with the taste signaling molecule α-gustducin [78,79]. TRPM4 has been identified as a homolog of MLSN (TRPM1) and is highly expressed in heart, kidney, prostate and colon and as a Ca^2+^ activated cation channel mediating membrane depolarization [80,81]. CAM binding sites in the C-terminus of TRPM4 are essential for regulating the sensitivity of direct Ca^2+^ dependent activation [76,82]. However, both TRPM4 and TRPM5 have a direct Ca^2+^ binding site on the intracellular side of the S1–S4 domain [83]. TRPM4, but not TRPM5, is inhibited by extracellular adenosine nucleotides (AMP, ADP, ATP) and its activity can be enhanced by PKC dependent phosphorylation of the TRP domain [84].

Increased TRPM4 expression correlates with decreased SOCE due to changes in the driving force for Ca^2+^ as has initially been demonstrated in prostate cancer cells [85]. A later report correlating the expression of TRPM4 with proliferation, cell cycle progression and invasion by colorectal cancer cells [86], but see also its role for cell spreading, migration and contractile behavior [87]. Within the heart, direct pathophysiological roles for TRPM4 that are likely independent from a concomitant inhibition of SOCE, are linked to smooth muscle depolarization and subsequent myogenic vasoconstriction [88]. Also, a recent review [8] provided the role of TRPM channels in several human diseases. In the context of SOCE regulation, the role of TRPM4 in the immune system is of particular interest. Bone marrow derived mast cells from TRPM4 knockout mice showed a greater release of leukotrienes and TNF-α as well as of histamine [89], but so far it has not been formally investigated to what extent these effects are dependent or independent from differential driving forces for Ca^2+^ influx from the outside. In T-lymphocytes, TRPM4-dependent alterations of cytokine production and altered Ca^2+^ oscillations as well as differential effects on NFATc1 localization [90,91], might also be caused by alterations of the electrical driving force. Still, the exact mechanism and/or the differential contribution of TRPM4 to immune responses remains to be fully understood and investigated for the response to differential T cell agonists. How ORAI and TRP channels interact has also been reviewed by Saul et al. [4]. As both TRPM4 and TRPM5 need Ca^2+^ for activation, a dual interaction with SOCE is likely with SOCE providing a source of Ca^2+^ and TRPM4/5 mediated depolarization subsequently providing negative feedback on SOCE mediated Ca^2+^ influx.

## 4. TRPM6 Channels

TRPM6 channels are involved in maintaining Mg^2+^ and Ca^2+^ homeostasis [13] and their activity is essential in the kidney and small intestine and in mammary epithelial cells and colon cells [12]. Mutations in the gene encoding TRPM6 cause hereditary disease of familial hypomagnesemia with secondary hypocalcemia [8]. TRPM6 channels can induce inward divalent cation currents when the intracellular Mg^2+^ levels are ~500 µM [12,14,92]. The permeability of TRPM6 to divalent cations is dependent on key acid residues present in its selectivity filter [12,93] and its ion conductance is estimated to be around 82–84 pS [12,94]. Molecular analysis based on amino acid sequences has revealed that TRPM6 and TRPM7 channels are close homologues, sharing a key feature of harboring the C-terminal serine/threonine protein kinase domain [3,10]. As shown in HEK-293 cells, TRPM6 specifically interacts with TRPM7 proteins forming complexes in the PM [95]. The TRPM6/7 complex has different biophysical properties compared to homomers of TRPM6, including its permeability to Ni^2+^, pore structure, inhibition by 2-APB, sensitivity to low (4–6) pH and conductance, ranging between 40 to 105 pS for TRPM7 and 56.6 pS for TRPM6/7 heteromeric form [3,12,94]. Additionally, the activation of TRPM6 channels is modified in the heteromeric form [92]. While TRPM6 homomers are inactive under basal Mg^2+^ levels, in the oligomeric form TRPM6 can be active after TRPM7 phosphorylation and lacks sensitivity to Mg^2+^ [92,94]. Despite the close homology of TRPM6 and TRPM7 channels they have different functions, with TRPM6 being involved in intestinal uptake and renal reabsorption of Mg^2+^, and TRPM7 regulating cellular Mg^2+^ homeostasis [12,14].

## 5. TRPM7 Channels

The TRPM7 presents a unique combination of an ion channel with a serine/threonine kinase and contributes to numerous physiological functions [96] (Figure 1). Besides being implicated in maintaining intracellular and systemic Mg^2+^ homeostasis [97,98,99,100,101,102], TRPM7 has been linked to cell motility, proliferation, differentiation, volume regulation, migration, and apoptosis [98,103,104,105,106,107,108,109,110,111,112,113,114,115,116,117,118,119]. This channel-kinase symbiosis paired with its ubiquitous expression gives it a central and non-redundant role in cellular processes. Its pathophysiology is broad being linked to neurodegenerative disorders, hypertension, and tissue fibrosis and to atypical immune responses [120,121,122,123,124,125,126,127]. As versatile as the function of TRPM7 is, so is its regulation by a variety of internal and external cellular factors. They range from intracellular cations, Mg-ATP, Cl^−^ and Br^−^ concentration, and intracellular pH to hydrolysis of the PIP_2_ [128,129,130,131]. The channel is constitutively active and conducts preferentially Zn^2+^, Mg^2+^ and Ca^2+^, and trace metals [103,115,132,133,134]. The constitutive active current is suppressed by intracellular levels of Mg^2+^ and Mg-ATP [103] and external Mg^2+^ acts as a permanent blocker of the pore [92,103,130,134]. These features led to the earlier designation of the native TRPM7 current as MagNuM (Mg^2+^-nucleotide-regulated metal ion current) [103,135] or MIC (Mg^2+^-inhibited cation current) [36]. TRPM7 permeability for Zn^2+^ is 4-fold higher than Ca^2+^ [134] and its deletion protects cells from Zn^2+^ and Ca^2+^ induced toxicity [136,137,138].

The kinase domain of TRPM7 belongs to the family of the atypical α-kinases [139] with the predisposition to phosphorylate serine and threonine residues in the context of an α-helix. It shows close homology to eukaryotic elongation factor-2 kinase (eEF2K), *Dictyostelium* myosin heavy chain kinases (MHCK) A-C and alpha-kinases 1-3 [140]. The first targets of the TRPM7 kinase were identified in vitro, including annexin A1 [141], PLC [142], and the MHCK A-C isoforms [105,143]. Novel mouse models made it possible to expand this diversity by the identification of native kinase substrates like SMAD2 [127,144]. Furthermore, the autophosphorylation of kinases supports target recognition and subsequent phosphorylation of substrates and appears not to be crucial for its catalytic activity [145,146,147]. Beside the necessity of Mn^2+^ or Mg^2+^ and Mg-ATP for the kinases activation and phosphorylation [146], the phosphorylation of some targets seems to be a Ca^2+^-dependent process [105,141,143]. This dependence suggests that channel activity somehow induces Ca^2+^ influx to support the kinase-target interaction. Additionally, the TRPM7 kinase can be cleaved by caspases to release the kinase domain, without losing its phosphotransferase activity from the channel and act on apoptotic signaling through the Fas receptor [119]. Interestingly, the cleaved kinase translocates to the nucleus affecting mRNA expression of TRPM7-dependent genes by modifying phosphorylation of serines and threonines on specific histone residues in a Zn^2+^ -dependent way [148].

The combination of a channel and a kinase poses a significant challenge to researchers when investigating the causes of pathophysiological processes. Since the discovery of TRPM7, the research has focused on the relationship between the channel and kinase activity and the physiological roles of the channel versus the kinase domain. This interplay between the TRPM7 channel and α-kinase activity affect each other, but the functional significance of this coupling is not clear and still the subject of ongoing investigation and debate. Reported inconsistencies result mainly from using a heterologous expression of TRPM7 mutants or by complete deletion of the kinase and the use of different tissue types [92,97,98,115,129,145,149]. A mouse model K1646R point mutation is introduced at TRPM7′s enzyme active site [96] opens the door to investigate the role of the kinase, uncoupled from the channel activity on physiological functions. Overall, it appears that the kinase activity of TRPM7 is not essential for the native channel function [96] and may play a more structural role in channel assembly or subcellular localization [97,98,145]. However, the kinase domain might still be important for Mg^2+^ levels ([Mg^2+^]_cyt_) mediated modulation of the TRPM7 channel itself, since the complete deletion of the domain increases [Mg^2+^]_cyt_ sensitivity of the channel [97,98,128] in contrast to inactive kinase point mutant K1646R [145]. Accordingly, the defect in Mg^2+^ homeostasis and TRPM7 current reduction is only found in heterozygous delta-kinase but not in K1646R mice [96,97,127].

## 6. TRPM8 Channels

The seminal discovery of TRPV1 by the Julius laboratory in 1999 [150] with the follow-up study of impaired nociception and pain sensation in mice lacking TRPV1 [151] and of the cold sensitive TRPM8 published in 2002 by the Julius [152] and Patapoutian [17] groups were groundbreaking. These studies were a critical step in recognizing that sensitization of primary afferent neurons via ion channels is responsible for detecting skin surface temperatures and thermal-related neuropathic pain. TRPM8 was originally cloned by screening cDNA isolated from trigeminal sensory neurons for their ability to respond to menthol and cold stimuli and from DRG neurons looking for novel sensory TRP channels [17,152], placing the TRPM subfamily at the center stage of thermal somatosensation [153]. However, the ability to sense changes in temperature is not restricted to a given TRP subfamily. The significance of these findings and the discovery of the touch-sensitive Piezo channels [154] won the discoverers Dr. David Julius and Dr. Ardem Patapoutian the 2021 Nobel Prize in Physiology [152,155]. TRPM8 channels are non-selective Ca^2+^-permeable channels exhibiting multi-gating mechanisms, being activated by innocuous cool to cold temperatures and regulated by crucial molecules such as Ca^2+^ and PIP_2_ [155,156]. The early work by the Latorre group identified the C-terminally located PIP_2_ sensing domain as one determinant of temperature sensitivity [157,158]. The permeability ratio between Ca^2+^ and Na^+^ ions (PCa/PNa) range from 0.97 to 3.2, with monovalent ions conductance series of Cs^+^ > K^+^ > Na^+^ [159]. Also, TRPM8 channels can depolarize cells and activate voltage-gated Na^+^ and Ca^2+^ channels [153], leading to increased ion influx. Basal cytosolic Ca^2+^ is required for TRPM8 activation by the cooling agonist, icilin, but not for menthol activity [24,160]. A major break-through in the understanding of TRPM8 modulation is based on several highly relevant reports published in 2017 to 2019 describing the cryo-electron microscopy structures of TRPM8. Herein, Yin et al. initially resolved the cryo-electron microscopy (Cryo-EM) structure of a full-length TRPM8 from the collared flycatcher [24]. Their structures revealed a complex layered architecture with the menthol binding site located within the voltage-sensor like domain [24,161]. This was followed by a report from the Julius group, using TRPM8 from another bird (*Parus major*) to more clearly resolve the transmembrane and selectivity filter domains of TRPM8 in the antagonist or Ca^2+^ bound configurations [153]. A second follow-up report from the Lee group in 2019 further revealed allosteric coupling between binding of PIP_2_ and cooling compounds and also revealed that intracellular Ca^2+^ is not necessary for cold- or menthol-dependent TRPM8 activation, however it is necessary for TRPM8 activation by icilin [162]. In contrast to competition for binding of PIP_2_ versus agonist binding in TRPV1 [163], TRPM8 has nearby but distinct binding sites for PIP_2_ and agonists and PIP_2_ is required as a cofactor for channel activation by cooling agonists and cold temperatures [153,164]. PIP_2_ itself might be sufficient to activate TRPM8, whereas depletion of basal PIP_2_ levels results in channel desensitization [153]. Endogenous ligands including testosterone, artemin and Pirt (phosphoinositide interacting regulator of TRP) protein have been also proposed as physiological ligands of TRPM8 channels [165]. The modulation of TRPM8 channels have been considered a promising approach to develop novel therapeutic tools for chronic pain and noxious cold sensitization [165] and have also been relevant in the treatment of cancer, neuropathic pain and inflammation [15,165].

**Table 1 cells-11-01190-t001:** TRPM1-8 channels ion influx characteristics including, enzymatic domain, gating, ion permeability, function, SOCE interaction and pharmacology.

TRPM Channels: Ion Influx Characteristics							
Name:	Enzymatic Domain	Gating	Ion Permeability	Function	SOCE Interaction	Pharmacology	References
TRPM1	No	Gα_o_ and Gβ_γ_ subunits of G-proteins	Divalent (Ca^2+^/Mg^2+^/Mn^2+^)	Skin Pigmentation Retinal Photoresponse	No	Activator(s): Pregnenolone Inhibitor(s): ↑[Zn^2+^]_cyt_	[51,52,58,60]
TRPM2	Yes (NUDT9-H)	ADP-ribose and Ca^2+^	Monovalent (Na^+^/K^+^/Cs^+^) Divalent (Ca^2+^/Mg^2+^/Ba^2+^)	Body Temperature Control Insulin/ROS/Immune Response	Yes (Indirectly)	Activator(s): ADP/ADPR analogues Inhibitor(s): Cacospongia/Scalaradial	[23,64,65,66,67]
TRPM3	No	G_i,q_-GPCRs Ca^2+^/CaM/MAPKs	Monovalent (Na^+^/K^+^/Cs^+^) Divalent (Ca^2+^/Mg^2+^/Ba^2+^)	Noxious Heat Sensation Glucose/Ca^2+^ Homeostasis	No	Activator(s): CIM0216/Pregnenolone Inhibitor(s): ↑[Mg^2+^]_cyt_/Primidone	[55,56,62]
TRPM4	No	Ca^2+^/CaM	Monovalent (Na^+^ > K^+^ > Cs^+^ > Li^+^ >> Ca^2+^/Cl^−^)	Myogenic Tone, Cardiac Conduction, Ca^2+^ Oscillation	Yes (Indirectly)	Activator(s): ↑[Ca^2+^]_cyt_ Inhibitor(s): AMP/ADP/ATP/DVT	[8,11,63,75,77,80]
TRPM5	No	Ca^2+^/CaM	Monovalent (Na^+^ ≥ K^+^ ≥ Cs^+^)	Taste, Insulin Secretion	No	Activator(s): ↑[Ca^2+^]_cyt_/PIP_2_/Rutamarin Inhibitor(s): TPPO	[3,74,77,81]
TRPM6	Yes (α-kinase)	PIP_2_/PLCγ	Mainly Mg^2+^/Ca^2+^ and other divalent (Ba^2+^/Zn^2+^/Mn^2+^)	Mg^2+^ Homeostasis Embryonic Development	No	Activator(s): ↓[Mg^2+^]_cyt_/EGF/Insulin Inhibitor(s): ATP/H_2_O_2_	[3,12,13,14,83,89]
TRPM7	Yes (α-kinase)	Phosphorylation PLCγ/Myosin IIA-C	Mainly Mg^2+^/Ca^2+^ and other divalent (Ba^2+^/Zn^2+^/Mn^2+^)	Mg^2+^ Homeostasis Cell Motility/Differentiation	Yes (Indirectly)	Activator(s): Naltriben/↓[Mg^2+^]_cyt_/PiP_2_ Inhibitor(s): NS8593/FTY720/WaxenicinA	[6,7,25,109,117]
TRPM8	No	Gα_q_-GPCRs/PIP_2_	Monovalent (Na^+^/K^+^/Cs^+^) Divalent (Ca^2+^/Mg^2+^/Ba^2+^)	Cold Skin Temperatures Thermal Neuropathic Pain prostate	Yes (Indirectly)	Activator(s): Menthol/Icilin/WS12 Inhibitor(s): AMTB/TCI2014/CPS-369	[24,152,154,159]

## 7. Discussion: The TRPM-SOCE Connection

The most evident connection between the TRPM channels and SOCE is represented by the TRPM7 channels. On the one hand, TRPM7, like SOCE, is implicated as a player in global Ca^2+^ levels and in mediating Ca^2+^ influx [104,141,166,167,168]. Moreover, Ca^2+^ influx through the channel seems to play a central role in controlling the recruitment of substrates, including annexin A1 and myosin II A heavy chain, to the TRPM7 kinase domain [105,141,143]. More importantly, recent reports highlighted the connections between TRPM7 and SOCE and their relevance to cell physiology [6,7,169]. Initial studies focused on the immune system cells because TRPM7 and SOCE play a significant role in immune cell development, activation, and the initiation of both innate and adaptive immune responses [33,116,170]. The deletion or pharmacological inhibition of TRPM7 reduces SOCE in DT 40 B-lymphocytes, which indicates a potential direct link between TRPM7 function and SOCE [6]. The observed reduction of CRAC currents (*I*_CRAC_) in the DT40 cells is neither due to membrane potential effects nor to indirect effects of K^+^ currents [6]. Furthermore, the authors exclude that TRPM7 channels are part of SOCE but considered these channels as SOCE regulators [6]. Rescue experiments in DT40 cells expressing kinase-dead (K1648R), or kinase-deficient mutant of TRPM7, provide evidence that regulation of SOCE takes place via its kinase domain [6]. Interestingly, TRPM7 participates in maintaining Ca^2+^ stores under resting conditions and contributes to ER store refilling after depletion [6].

An ensemble between SOCE and TRPM7 appears to support the intracellular Ca^2+^ balance under resting conditions and after activation of the Ca^2+^ signaling cascade [6]. In line with these findings, a second study used pharmacological (modulators) and molecular (siRNA) approaches to examine SOCE/TRPM7 connections in primary enamel forming cells (ameloblasts) and in the enamel cell line LS8 cells [7]. Especially during ameloblast differentiation, the TRPM7 kinase plays a role by phosphorylating the cAMP response element binding (CREB) protein [22]. The use of naltriben as a TRPM7 activator [7,171] enhances SOCE in rat-derived primary ameloblasts from the secretory and maturation stages [7]. Pharmacological suppression of TRPM7 pore does not decrease SOCE and excludes TRPM7 as a component of SOCE in these cells, supporting the non-critical role of TRPM7 channels on SOCE in ameloblasts [7]. Moreover, the activation of TRPM7 with naltriben in LS8 cells lacking both *ORAI1* and *ORAI2* (sh*ORAI1-2*) failed to increase cytosolic Ca^2+^ levels [7]. These findings are supported by data on primary ameloblasts of *Stim1/2^K14cre^* mice lacking *Stim1* and *Stim2*, which had not been previously reported. Figure 2 shows that whereas naltriben enhances the SOCE peak in ameloblasts of wild type mice, the activation of TRPM7 in ameloblasts of *Stim1/2^K14cre^* mice, which show nearly abolished SOCE [28], failed to show any changes in Ca^2+^ influx. This supports the notion that the potentiating function of TRPM7 on SOCE likely requires the previous activation of SOCE, and that TRPM7 is not able to compensate for the lack of SOCE. Furthermore, additional rescue experiments of the TRPM7-KO-mediated phenotype with the inactive kinase mutant (K1648R) or hTRPM7 ∆-kinase could address the involvement of the channel or kinase in this potentiating function [6,7], as it has already been studied in B-lymphocytes.

Overall, additional studies seem to be required to elucidate the role of TRPM7 or its kinase in enamel cells and beyond. If the TRPM7 kinase is the regulatory portion in this context, the kinase inactivated mouse model should unmask its role in Ca^2+^ signaling. Inactivation of the TRPM7 α-kinase in a mouse model leads to splenomegaly with increased splenocytes numbers [172], but unaltered T cell subsets distribution in the spleen [127]. TRPM7 activity in murine splenic T cells is comparable in wild-type TRPM7 and KD mutant, highlighting the dispensability of the kinase function for ion conduction again [96,145]. Also, the SOCE magnitude tends to be larger in cells from KD mutant mice, and the initial slope is significantly increased, suggesting a potentiated Ca^2+^ influx through ORAI channels. The kinase deactivation causes potentiation of Ca^2+^ signals in resting conditions while a reduction occurs in the activated cells. The reasons can be multi-layered, including differences in metabolic stages of the cells or altered SOCE expression patterns within subpopulations of T cells [173,174,175].

Recent evidence highlighted the TRPM7 α-kinase domain as a further indirect modulatory player generating Ca^2+^ signals via SOCE [6]. So far, there is a lack of evidence supporting the active participation of TRPM7 in SOCE, at least in the cells studied so far [6,7]. This raises the next question: how does TRPM7 indirectly exert these regulatory abilities on the SOCE pathway? The obvious answer would be its ability to phosphorylate targets directly or indirectly involved in the SOCE pathway. The reversible phosphorylation of proteins catalyzed by kinases is central in regulatory mechanisms of signal transduction [128]. Several studies have already identified ORAI and STIM proteins as targets for kinases, leading to alterations of Ca^2+^ entry [176,177,178,179]. In addition, STIM and ORAI regulatory proteins including SOCE-associated regulatory factor (SARAF), CRACR2A, GOLLI proteins, caveolin and septin [180,181,182,183] may also be under the influence of kinases. Another interesting point is the involvement of TRPM7 to form the myosin II motor protein, bundle actin filaments, and build the actomyosin cytoskeleton network. The MHC-II phosphorylation by TRPM7 is Ca^2+^-dependent [105], and it relies on STIM1-mediated Ca^2+^ entry since STIM1 deletion abolished actomyosin formation [184]. These reciprocal interactions between TRPM7 and SOCE may require tight balance to maintain proper cell signaling and functioning.

Additional SOCE/TRPM links are provided by TRPM2 channels. Liu et al. reported that in irradiated salivary glands, TRPM2 channels are activated resulting in elevated mitochondrial Ca^2+^ and ROS which subsequently induced caspase-3 cleavage of STIM1 and loss of SOCE [73]. While these data strongly suggest a cause-effect association between SOCE and TRPM2, this is an indirect link as it does not suggest a direct interaction between TRPM2 and any of the SOCE components [73]. TRPM8 channels have also been proposed to antagonize the degree of SOCE. The downregulation of TRPM8 in pulmonary smooth muscle cells correlates with increased SOCE, and the application of icilin causes suppression of SOCE. However, the link is less clear and, thus far, likely indirect [185]. TRPM8 (and many other channels) activity can be influenced by alterations in Ca^2+^ and PIP_2_ levels [153], and are also affected by the activity of the GPCR Gα_q_-subunit which could disrupt the PLC-PIP_2_ signaling cascade [186] and therefore might indirectly affect SOCE. A general note of caution is that activation of cation selective channels with a sufficient ion flux capacity and high Na^+^ > Ca^2+^ permeability ratios will lead to membrane depolarization, which might indirectly reduce influx through STIM2 mediated pre-coupled ORAI channels, thus potentially lowering basal Ca^2+^.

## 8. Concluding Remarks

Since the identification of ORAI1-3 and STIM1/2 as key components of SOCE, several reports have suggested that members of the TRPM family are associated with SOCE. However, the available evidence at present does not support the consideration that TRPM channels are intrinsic components of SOCE. Nonetheless, at least two members of the TRPM family (TRPM2 and TRPM7) can modulate SOCE, albeit such modulation appears to be primarily indirect involving either the phosphorylation of SOCE components via the enzymatic domain of TRPM7, or via mitochondrial Ca^2+^ accumulation and ROS generation to degrade STIM1. Yet there are several gaps in understanding the nature of these modulatory functions, particularly evident in the case of TRPM7. Additional work is required to better discern the synergy between TRPM members and SOCE and its impact on the Ca^2+^ signaling cascade.

## Figures and Tables

**Figure 1 cells-11-01190-f001:**
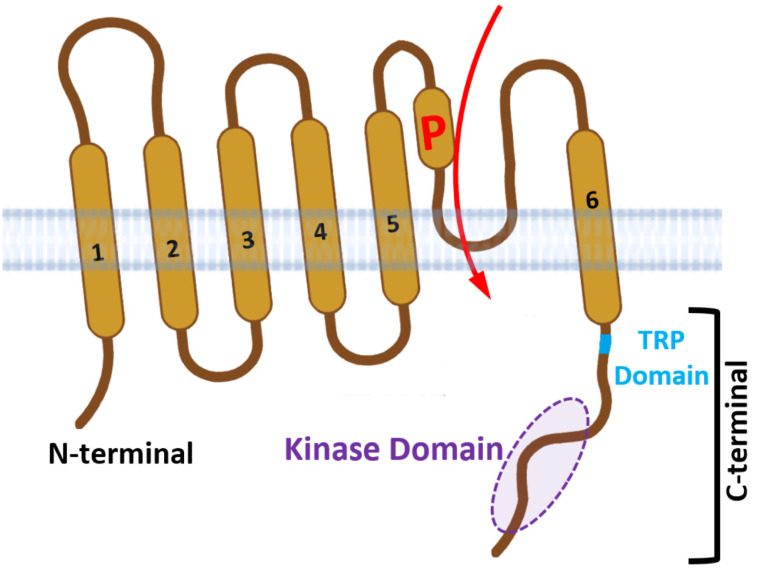
**Structural architecture of TRPM7.** TRPM7 channels are formed by six helical transmembrane domains (TM1-TM6). The channel pore (P) of TRPM7 is located between TM5 and TM6. TM1 contains the N-terminal region and TM6 harbors the serine/threonine kinase domain. The N’ and C’- terminal regions are in the cytosol.

**Figure 2 cells-11-01190-f002:**
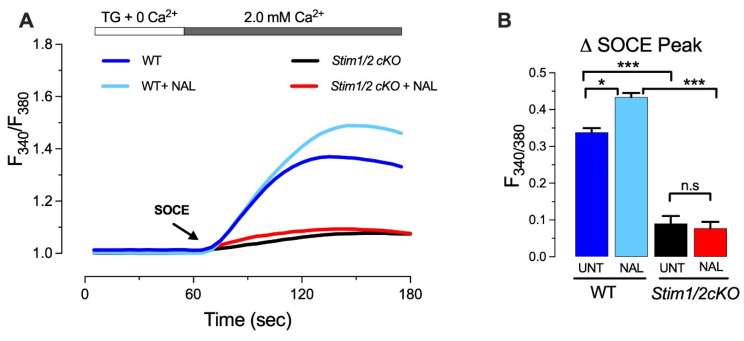
**TRPM7 stimulation does not elicit Ca^2+^ influx in SOCE-deficient ameloblasts**. (**A**) Representative original traces of [Ca^2+^]_cyt_ transients in ameloblasts of *Stim1/2^K14cre^* mice (*Stim1/2cKO*) and controls (WT) ameloblasts. The ameloblasts of *Stim1/2^K14cre^* mice were isolated as reported (28). SOCE was measured after pre-incubation with thapsigargin (20 min, 1 μM), followed by perfusion with a Ca^2+^-free Ringer’s solution (60 s) before simultaneous re-addition of 2.0 mM extracellular Ca^2+^ or with 2 mM Ca^2+^ and the TRPM7 agonist naltriben (NAL,100 μM). (**B**) Quantification of the SOCE peak. Data were analyzed by one-way ANOVA followed by Tukey’s multiple comparison post-hoc test. *****
*p*< 0.05, *** *p* < 0.001, n.s., non-significant.

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
