# Peer review of "On the Connections between TRPM Channels and SOCE"

_cells, 2022, doi:10.3390/cells11071190_

Round 1
Reviewer 1 Report
Summary
The review focuses on the function and biophysical properties of TRPM family members and their interaction with store-operated calcium entry components in various tissues.
General comments:
Many reviews are cited here but some cover essentially identical topics. For example, several of the references 1, 30, 31, 32, 34, 41, 44 can be omitted to reduce redundancy. The same goes for the TRPM reviews: consider deleting the older ones. It would benefit the reader if original articles are cited more often in addition to review articles.
Specific
Line 42: since ref. 7 preceded 6 by three years, their sequence in the bibliography should be reversed if possible.
Line 60: it would be useful to mention here “eEF2K-like” as an alternative way of naming alpha kinases. Although it was initially thought that the residues phosphorylated by these kinases were in alpha helices, this is not longer the case. You might want to mention on page 5 the old names of TRPM7 and 6, ChaK1 and 2 (channel kinase 1, 2).
Line 74: Table 1 appears to be limited to mammalian TRPM, its title should reflect that. There are several errors in this table, listed below-
Increased cytosolic Mg2+ is not an activator of TRPM6 or TRPM7 channels, it is an inhibitor but a kinase activator, please clarify that. Also, like other melastatin subfamily members, TRPM7 and 6 conduct monovalents.
There are other known inhibitors of TRPM7 like 2-APB and SK96395 that should be added to the list. Please correct FTY20 to FTY720.
It has been known for some time that phosphorylation is not the mechanism of TRPM7 gating: the channel activity of TRPM7 kinase-dead mutants is normal.
TRPM1: Galpha and Gbeta/gamma are subunits not of GPCRs but G-proteins. They are considered downstream of GPCR activation.
TRPM2: correct anologues to analogues.
TRPM3: Gi is a type of alpha subunit, G alpha i.
TRPM4 has other well-known inhibitors like 9-phenanthrol.
TRPM4: Correct oszillation to oscillation.
TRPM5 is activated downstream from the G protein alpha-gustducin. TRPM5 was originally characterized in rodent taste cells where gustducin is expressed.
Generally, it is thought that all TRPM channels require PIP2 and they all conduct monovalent cations such as Na+ and K+.
Line 61 and 228: a recent publication by Mellott et al, Pflugers Archiv, showed that the outward TRPM7 current has a clear function in cellular Mg2+ loading and that inhibition or deletion of TRPm7 does not protect against Zn2+ induced T-cell toxicity.
Line 88: Orai1-3 biophysical properties were characterized in detail in 2008 by four groups (Penner, Cahalan, Putney, Romanin), those papers should be included in references.
Line 143: TRPM2 ADPRase enzymatic domain is vestigial in the human and other mammals. But it is functional in invertebrate orthologs (see Iordanov et al 2019, eLife). This information is important and should be added.
Line 152: instead of changes in intracellular Ca levels, simply write ‘increases’.
Line 155: the wrong Prawitt reference [73] is cited: the Mtr1 Prawitt et al. papers were published in 2000.
Line 166: add parentheses before ‘but’.
Line 174: insert ‘on’ after ‘dependent’.
Line 184: reference [86] is too speculative and best omitted.
Figure 1: In panel A, in order to show that this calcium elevation is store dependent, the authors may want to include a control experiment in WT cells of preincubation in 0 Cao without thapsigargin addition (no store depletion), followed by 2 mM Ca2+. Was the store calcium release transient observed in WT and STIM KO ameloblasts during acute Tg addition in the absence of calcium? These questions are of importance particularly in view of the recent paper by Kádar (IJMS 2021) on TRPM7-mediated calcium transport in ameloblasts that has not been addressed.
Line 262-264: Studies from other groups, on the other hand, suggest that complete deletion of the kinase domain results in non-functional TRPM7 channels, which would explain reduced current magnitudes in heterozygous kinase-deleted mouse cells.
Line 283-284: Ref. 3 is a review primarily discussing TRPM protein structures rather than permeation properties and [153] discusses TRPM8 agonist compounds. It would be useful to cite the original papers demonstrating Na+ permeation. E.g. Lückhoff’s 2007 JBC paper addressed TRPM8 permeation properties.
Lines 317, 332: Please check refs. 165, 92, 167 for relevance in this context.
Line 368-369: the authors are probably referring to the potentiated SOCE in quiescent lymphocytes, as their next sentence implies. Nevertheless, I recommend mentioning ‘resting’ or ‘quiescent’ cells here as well, for more clarity.
Reviewer 2 Report
Bomfim et al, “On the connections between TRPM Channels and SOCE”
This is a very timely and well written review to connect known studies linking SOCE to TRPM channels. As such the authors have nicely introduced the TRPM family of ion channels and discussed later the literatures connecting them to SOCE. While in its present form the review is very comprehensive and informative, I would just recommend the following two minor things that would tremendously benefit a reader if added to the manuscript:
- In Table 1 describing differences between different TRPM channels, can the authors add one section describing known channelopathies associated with each of them? These have been mentioned in the text but its summary with the table will be very useful.
- A schematic of TrpM or specifically TrpM7 onto which the authors focus most during their discussion will also make a beautiful addition to the review, especially for novices in the field.
Overall it is a very well written and informative review.
Reviewer 3 Report
This review by Bomfim et al addresses the question of interdependence of the TRPM and store-operated channels. The authors analyse a good selection of relevant papers; however, some papers are misquoted, and the manuscript would benefit from some editing. Please see specific comments in the annotated attached file.
